# High Antibiotic Resistance of *Helicobacter pylori* and Its Associated Novel Gene Mutations among the Mongolian Population

**DOI:** 10.3390/microorganisms8071062

**Published:** 2020-07-16

**Authors:** Dashdorj Azzaya, Boldbaatar Gantuya, Khasag Oyuntsetseg, Duger Davaadorj, Takashi Matsumoto, Junko Akada, Yoshio Yamaoka

**Affiliations:** 1Department of Environmental and Preventive Medicine, Faculty of Medicine, Oita University, Yufu City, Oita 879-5593, Japan; azzaya2000@gmail.com (D.A.); tmatsumoto9@oita-u.ac.jp (T.M.); akadajk@oita-u.ac.jp (J.A.); 2Department of Gastroenterology, Mongolian National University of Medical Sciences, Ulaanbaatar 14210, Mongolia; medication_bg@yahoo.com (B.G.); oyuntsetseg.kh@mnums.edu.mn (K.O.); davaadorj55@hotmail.com (D.D.); 3Endoscopy Unit, Mongolia-Japan Teaching Hospital, Mongolian National University of Medical Sciences, Ulaanbaatar 250573, Mongolia; 4Department of Medicine, Gastroenterology and Hepatology section, Baylor College of Medicine, Houston, TX 77030, USA

**Keywords:** *Helicobacter pylori*, antibiotic resistance, whole genome sequencing, resistance mutation, next-generation sequencing, Mongolia

## Abstract

Mongolia has a high prevalence of *Helicobacter pylori* infection and the second highest incidence of gastric cancer worldwide. Thus, investigating the prevalence of antibiotic resistance and its underlying genetic mechanism is necessary. We isolated 361 *H. pylori* strains throughout Mongolia. Agar dilution assays were used to determine the minimum inhibitory concentrations of five antibiotics; amoxicillin, clarithromycin, metronidazole, levofloxacin, and minocycline. The genetic determinants of antibiotic resistance were identified with next-generation sequencing (NGS) and the CLC Genomics Workbench. The resistance to metronidazole, levofloxacin, clarithromycin, amoxicillin, and minocycline was 78.7%, 41.3%, 29.9%, 11.9% and 0.28%, respectively. Multidrug resistance was identified in 51.3% of the isolates investigated which were further delineated into 9 antimicrobial resistance profiles. A number of known antibiotic resistance mutations were identified including *rdxA*, *frxA* (missense, frameshift), *gyrA* (N87K, A88P, D91G/N/Y), 23S rRNA (A2143G), *pbp1A* (N562Y), and 16S rRNA (A928C). Furthermore, we detected previously unreported mutations in *pbp1A* (L610*) and the 23S rRNA gene (A1410G, C1707T, A2167G, C2248T, and C2922T). The degree of antibiotic resistance was high, indicating the insufficiency of standard triple therapy in Mongolia.

## 1. Introduction

The discovery of *Helicobacter pylori* revolutionized pathological concepts and therapeutic approaches to upper gastrointestinal diseases. *H. pylori*, which is a major cause of chronic gastritis, peptic ulcer, mucosa-associated lymphoid tissue (MALT) lymphoma, and gastric cancer (GC) [1,2,3] was classified as a class I carcinogen in 1994 [4]. At least 1 million new cases of GC were diagnosed in 2018, and GC is the third leading cause of cancer death worldwide [5]. Mongolia, which is bordered by Russia and China, is one of the least densely populated countries in the world. The population of Mongolia has a high prevalence (76%) of *H. pylori* infection [6] and 80% among dyspeptic patients [7]. Moreover, Mongolia has the second-highest incidence of GC with the highest global mortality rate [8]. According to the age-standardized rate (ASR) of GC per 100,000 Mongolian population, the highest rate (80.7) has been reported in the Uvs Province (West), the least (11.2) in Khentii Province (East), and a national average of 32.4 [8]. Therefore, the effective treatment of *H. pylori* infection requires eradicating the pathogen and preventing its associated diseases such as GC [2].

Standard triple therapy for *H. pylori* infection comprises amoxicillin and clarithromycin or metronidazole with acid inhibitors. However, the global cure rate achieved using standard triple therapy has declined to less than 80% [9,10]. The main explanation of this unacceptable cure rate is antibiotic resistance, particularly to clarithromycin, metronidazole, or both [11]. The World Health Organization declared that clarithromycin-resistant *H. pylori* is a high priority bacterium in the same tier (critical, high, or medium) as methicillin-resistant *Staphylococcus aureus* [12]. In China [13], Japan [14], and Sweden [15], the rates of resistance to clarithromycin were 31%, 38.5%, and 1.5%, respectively. Resistance to clarithromycin is generally associated with point mutations in the 23S rRNA gene [10]. Metronidazole resistance varies among countries; however, in the Asian-Pacific region it was 44% [11].The mechanism of the metronidazole resistance is complex, involving mutations in *rdxA*, which encodes oxygen-insensitive NADPH nitroreductase [16], and *frxA*, which encodes NAD(P)H-flavin oxidoreductase [17]. Furthermore, resistance to levofloxacin, which is often used as rescue therapy, is increasing. Levofloxacin resistance is frequently associated with GyrA and GyrB [18]. The average global rate of tetracycline resistance is >10% [19], and resistance is mainly associated with the substitution AGA_926-928_ -> TTC in the 16S rRNA gene [20].

There are no specific guidelines for treating *H. pylori* infection in Mongolia. Thus, infection has been eradicated according to the international consensus [2,21]. Only one study reports the prevalence of antibiotic resistance in Mongolia [22]. This study was limited because it was conducted in Ulaanbaatar, did not investigate the main antibiotics such as levofloxacin, and did not investigate the genetic mechanisms of antibiotic resistance in detail [22]. The antibiotic susceptibility pattern of regional populations is essential for selecting an effective treatment regimen. Therefore, we aimed to investigate the rates of antibiotic resistance of *H. pylori* to amoxicillin, clarithromycin, metronidazole, levofloxacin, as well as to minocycline, which is a second-generation tetracycline derivative. Moreover, we conducted whole genome sequencing (WGS) to identify gene mutations associated with the antibiotic resistance of *H. pylori* and their distribution within different regions of Mongolia.

## 2. Materials and Methods

### 2.1. Study Population and Sampling

We conducted a cross-sectional study of dyspeptic patients from November 2014 to August 2016. We enrolled patients >16 years old willing to undergo an upper gastrointestinal (GI) endoscopic examination. We excluded patients with a history of partial or total gastrectomy, treatment with H_2_-receptor blockers or proton pump inhibitors within 4 weeks before the study, and previous eradication of *H. pylori* infection. The study areas were selected according to location and their ASRs of GC in 2012. Mongolian demographic data (ASRs) are as follows: Ulaanbaatar city (central Mongolia; 31.3), Uvs Province (West, 80.7), Khuvsgul Province (North, 37.0), Umnugovi Province (South, 21.5), and Khentii Province (East, 11.5).

### 2.2. H. pylori Isolation and DNA Sequencing

#### 2.2.1. Isolation and Culture of *H. pylori*

During upper GI endoscopy, two biopsy specimens were taken from the antrum, approximately 3 cm from the pyloric ring. One specimen was used for the rapid urease test (RUT) (MON-HP, Mongolian National University of Medical Sciences, Ulaanbaatar, Mongolia). The other was immediately placed at −20 °C and stored at −80 °C at each location, kept on dry ice during transfer to Ulaanbaatar city, kept at −80 °C at Mongolian National University of Medical Sciences, transferred with dry ice to Oita University, and stored again at −80 °C. To culture *H. pylori*, the specimens were homogenized and used to inoculate a commercial *H. pylori* selective plate (Nissui Pharmaceutical Co. Ltd., Tokyo, Japan). The plates were incubated for 3–7 days at 37 °C under microaerophilic conditions (10% O_2_, 5% CO_2_, and 85% N_2_). Small, purple colonies typical of *H. pylori* were subjected to Gram staining and tested for oxidase, catalase, and urease activities.

#### 2.2.2. DNA Extraction and WGS

The genomic DNAs of *H. pylori* isolates were extracted using a QIAamp DNA Mini Kit (QIAGEN, Hilden, Germany) according to the manufacturer’s instructions. DNA quality was assessed using a Quantus Fluorometer (Promega, Madison, WI, USA). WGS was performed using an Illumina MiSeq (Illumina, San Diego, CA, USA) platform, and the DNA library was prepared using the Nextera XT DNA sample kit (Illumina) to generate 2 × 300 bp paired-end reads.

### 2.3. Antibiotic Susceptibility Tests and Detection of Genotypic Determinants of Antibiotic Resistance

#### 2.3.1. Antibiotic Susceptibility Test

Antibiotic susceptibility was detected using a serial two-fold agar dilution assay to determine the minimum inhibitory concentrations (MICs) of amoxicillin, clarithromycin, metronidazole, levofloxacin, and minocycline (Wako Pure Chemical Industry, Osaka, Japan) according to the guidelines of the Clinical and Laboratory Standards Institute (CLSI) (Wayne, PA, USA). Briefly, bacteria were subcultured on Mueller-Hinton II Agar medium (Becton Dickinson, Sparks, MD, USA) supplemented with 5% defibrinated horse blood. The bacterial suspension was adjusted to OD_600_ = 0.1, and a 48-pin inoculator was used to inoculate the culture plate (1 μL per spot, approximately 10^4^ colony forming units [CFU] of bacteria). *H. pylori* strain 26695 was used as a control strain. MICs were judged according to the presence or absence of growth at the spots at the lowest concentration of antibiotic, followed by checking growth at the spots using 1:1 dilutions after 72-h incubation. Resistance or sensitivity to antibiotics was judged according to the guidelines of the European Committee on Antimicrobial Susceptibility Testing (EUCAST; http://www.eucast.org/). The clinical breakpoints of MICs indicating antibiotic resistance are >0.125 mg/L, amoxicillin; >0.5 mg/L, clarithromycin; >8 mg/L, metronidazole; >1 mg/L, levofloxacin; and >1 mg/L, minocycline. Duplicate agar dilution assays were repeated 2–3 times.

#### 2.3.2. Detection of Genotypic Determinants of Antibiotic Resistance

We determined the whole genome sequences of 74 Mongolian *H. pylori* isolates. The filtering and trimming reads were processed using CLC Genomics Workbench software v11.0.2 (Qiagen) following Illumina’s recommendations. We selected samples with >80% reads with quality ≥Q30. After quality checking, the sequences were assembled using the default parameters of the CLC Genomics Workbench. Sequence reads were mapped against the reference strain 26695 (GenBank accession number AE000511.1). The complete nucleotide sequences of *pbp1A*, *rdxA*, *frxA*, *gyrA*, 23S rRNA, and 16S rRNA, likely involved in antibiotic susceptibilities, were extracted using the BLAST algorithm implemented in the CLC Genomics Workbench and compared with the reference strain. Nucleotide sequence datasets and inferred amino acid sequences were aligned and visually analyzed using the CLC Genomics Workbench and MEGA v7 [23].

#### 2.3.3. Statistical Analysis 

SPSS statistics version 20.0 (SPSS Inc., Chicago, IL, USA) was used for statistical analysis. The chi-square test or Fisher’s exact test were used. *p* < 0.05 indicates a significant difference.

#### 2.3.4. Nucleotide Sequence Accession Number

All nucleotide sequences analyzed in this study were deposited in the DNA Data Bank of Japan under accession number ID: LC567134-LC567141, LC567329-LC567379, LC568549-LC568586.

### 2.4. Ethics

Written informed consent was obtained from all subjects, and the Ethics Committees of the Ministry of Health (acceptance number No. 03, 11 September 2015), Ethics Committee at Mongolian National University of Medical Sciences (N13-02/1A, 11 June 2015) and by Ethics Committee at Oita University Faculty of Medicine (Yufu, Japan) (P-12-10, 17 January 2013, and No. 1660, 19 July 2019) approved this study.

## 3. Results

We isolated 361 *H. pylori* strains from biopsies of individuals who participated in gastric examinations at the locations as follows: Ulaanbaatar city (*n* = 124), Uvs Province (*n* = 28), Khuvsgul Province (*n* = 35), Khentii Province (*n* = 90), and Umnugovi Province (*n* = 84). Patients’ mean age was 44.3 ± 13.4 (mean ± SD) years, and 73.1% (264/361) were women (Table 1). The frequency of resistance to metronidazole was highest (78.7%), followed by levofloxacin (41.3%), clarithromycin (29.9%), and amoxicillin (11.9%), and one strain was resistant to minocycline (0.28%) (Table 1 and Figure 1). Isolates from females were more resistant to metronidazole (*p = 0.003*). Age was not significantly associated with resistance.

The distribution of antibiotic resistant rates according to locations are shown in Table 2. Resistance to clarithromycin was highest in Uvs Province (50%) and the lowest in Khuvsgul Province (17.1%) (*p* = 0.034). The highest resistance rate to amoxicillin was in Uvs Province (25%) and the lowest in Umnugovi Province (7.1%) (*p* = 0.043). The resistance rates to metronidazole, minocycline, and levofloxacin were not significantly associated with geographical location.

Table 3 shows the multidrug resistance (MDR) patterns. Only 11.9% (43/361) of isolates were susceptible to five antibiotics, and 51.3% (185/361) were MDR-associated, with nine distinct MDR profiles. Among the 133 strains resistant to only one drug, the majority were resistant to metronidazole (29.4%). The highest frequency of double-drug resistance was to metronidazole and levofloxacin (18.3%), followed by metronidazole and clarithromycin (9.1%). All triple-drug resistance profiles included resistance to amoxicillin, metronidazole, or clarithromycin; and the highest rate was for the combination of clarithromycin, metronidazole, and levofloxacin (43 [11.9%]). Ten (2.8%) strains were resistant to four antibiotics.

### 3.1. Genotypic Determination of Antibiotic Resistance

To understand the basis for the high antibiotic-resistance phenotypes in Mongolia, we determined the full sequences of 74 (20%) strains, including 10 strains that were sensitive to each of the five antibiotics (Table 3). We analyzed the annotations of genes associated with antibiotic resistance to identify putative mutations encoding antibiotic resistance.

#### 3.1.1. Amoxicillin Resistance

We assessed genetic variants in the gene (*pbp1A)* encoding penicillin binding protein 1A, including 12 amoxicillin-resistant and 62 susceptible strains. Mutations in or adjacent to penicillin-binding protein motifs (PBP motifs) SXXK_368-71_, SXN_433-5_, and KTG_555-7_ of *pbp1A* confer amoxicillin resistance [24,25]. Most (*n* = 60) amoxicillin-sensitive strains did not carry these mutations (Figure 2). Intriguingly, N562Y, which is adjacent to motif KTG_555-7_, was significantly associated with amoxicillin resistance (*p* = 0.00001). This mutation was detected in six (50%) amoxicillin-resistant strains, including strains with a high MIC (0.5 or 1 μg/mL). Further, we noted other substitutions in PBP1A, which were only detected in amoxicillin-resistant strains. Strain Kh 93 carried the mutation T556S in the PBP motif, KTG_555-7._ Strain Kh 130 carried the substitution V374L, and strain UB214 carried the triple substitution V374L, S414N, and I450V. S414N and I450V are adjacent to the motifs SXXK_368-71_ and SXN_433-5_, respectively. The MICs of strains Kh130 and UB214 were the same, indicating that V374L is likely responsible for resistance, whereas S414N and I450V had minor or no effect on resistance. Interestingly, strain Kh 56 carried the substitution S414R as well as a unique nonsense mutation that generated the stop codon L610. The S414R mutation was detected in two sensitive strains, which was therefore unrelated to amoxicillin resistance. In contrast, the nonsense mutation carried by strain Kh56 was associated with resistance. Two amoxicillin-resistant strains (16.7% of resistant strains) did not carry these mutations in *pbp1A*.

#### 3.1.2. Clarithromycin Resistance 

We investigated the mutations in the 23S rRNA gene, particularly in domain V, of 19 clarithromycin-resistant and 55 sensitive strains. Point mutations in the peptidyl transferase region of domain V of 23S rRNA are associated with clarithromycin resistance [10]. Twelve types of point mutations were detected among the 19 clarithromycin-resistant strains. The known mutation, A2143G, was detected in 13 (68.4%) resistant strains and was significantly related to clarithromycin resistance (*p* = 1.99 × 10^−11^). Five clarithromycin-resistant isolates had the double mutations A2143G + C1604T, A2143G + C1625T, A2143G + C1669T, A2143G + C1848T, and A2143G + G2921A, which all contained A2143G (Figure 3). However, these additional mutations were not associated with increased MICs compared with the single mutation in A2143G, suggesting it had a minor or no effect on resistance. However, we were surprised to detect the mutation A2143G in two clarithromycin-sensitive strains. Moreover, other nucleotide substitutions in the 23S rRNA gene at A1410G (strain Uvs 147), C1707T (strain UB 130), A2167G (strain Ke 136), and C2922T (strain UB 183), were only detected in clarithromycin-resistant strains. Strain UB 221 carried the double point mutations C2248T + G2287A. Interestingly, the highly resistant strain Go 155 did not carry a mutation in the 23S rRNA domain V. Furthermore, most clarithromycin-sensitive strains (53/55, 96.4%) lacked mutations in domain V.

#### 3.1.3. Metronidazole Resistance 

Mutations in *rdxA*, which encode oxygen insensitive NADPH nitroreductase, and inactivation of *frxA*, which encodes NADPH flavin oxidoreductase, are associated with resistance to metronidazole [16,17]. Fifty-eight (78.4%) strains were resistant to metronidazole (Figure 4), and among the resistant strains, 30/58 (51.7%) carried missense mutations and 6/58 (10.3%) carried a nonsense mutation in *rdxA* (Figure 4). Six strains (10.3%) had the deletion, and two strains (3.4%) carried a frameshift mutation in *rdxA.* Twenty-five strains carried missense mutations in *frxA*, and the frameshift mutation was predominantly detected in *frxA* (24/58) compared with *rdxA*. Two strains carried a nonsense mutation, and *frxA* was deleted from one strain. The sequences of two strains were incomplete. Interestingly, the two strains that were moderately resistant to metronidazole (MIC 32–64 μg/mL) did not carry mutations in *rdxA* or *frxA*.

#### 3.1.4. Levofloxacin Resistance

Twenty-five (33.8%) sequenced strains were resistant to levofloxacin. Fluoroquinolone resistance is explained by mutations in the quinolone-resistance determining region (QRDR) of *gyrA* [26]. Amino acid substitutions at positions N87K (*p* = 0.00009), D91G (*p* = 5.1 × 10^−9^), D91N (*p* = 0.00002), and D91Y (*p* = 0.001) were significantly associated with levofloxacin resistance. Most strains with MICs ranging from 8 to 16 carried the mutation N87K, and one strain had an amino acid substitution in codon Ala_88_ -> Proline (A88P). Three levofloxacin-resistant strains did not carry a mutation in codons Asn87 to Asp91. All the levofloxacin-susceptible strains carried a mutation in the QRDR (Figure 5 and Appendix A).

#### 3.1.5. Minocycline Resistance

Minocycline serves as an alternative treatment against *H. pylori* infection. Resistance to tetracycline or its derivatives such as minocycline is explained by the triple base-pair substitution AGA_926-928_->TTC in the 16S rRNA gene [20]. The single isolate was resistant to minocycline (MIC = 2 μg/mL) carried a substitution in codon 928 (adenine for cytosine) in the 16S rRNA gene, which was not detected in the sensitive strains.

## 4. Discussion

Here we determined the antibiotic susceptibility rates of five antibiotics of 361 *H. pylori* strains isolated in five regions of Mongolia. One-fifth (74 strains) were selected for WGS, which allowed us to identify variants of genes associated with resistance to five antibiotics and to investigate the putative mechanisms.

Resistance to amoxicillin, which is a key component of triple therapy, is relatively low, 0% or <5% worldwide [10]. Very few Asian countries, including India and Pakistan, have a high incidence of amoxicillin-resistant *H. pylori* [11]. Moreover, 11.9% of Mongolian strains analyzed here were resistant to amoxicillin (as much as 25% in Uvs Province) (Table 2). Our previous randomized clinical trial and prevalence studies of Ulaanbaatar city found high resistance rates to amoxicillin, 8.4% and 23.0%, respectively [22,27], suggesting that Mongolia serves as model to investigate the mechanism of high amoxicillin resistance. A study of antibiotic use in Mongolia conducted in 2018 found that beta-lactam antibiotics are the most frequently used (58.9%), and amoxicillin alone represented 31.1% of the most frequently used orally-administered drugs [28].

Amoxicillin resistance is caused by mutations in or adjacent to the motifs SXXK_368-71_, SXN_433-5_, and KTG_555-7_ of *pbp1A* [29]. Here we show that the known mutation N562Y, which is adjacent to the PBP motif KTG_555-7_, was significantly related to resistance. Further, V374L and T556S mutations were detected here only in the resistant strains. One strain with the mutation V374L carried the previously unknown S414N and I450V substitutions, and another strain carried a single mutation V374L. However, both had the same MICs, indicating that the S414N and I450V substitutions likely do not play a major role in resistance. However, V374L may play a role in the resistance of Mongolian strains. For example, Val_374_→Leu confers amoxicillin resistance compared with the amino acid sequences of PBP1A of 77 each of amoxicillin-sensitive and -resistant strains [30].

Intriguingly, we discovered that strain Kh 56 carried the S414R mutation with a unique early stop codon at position 610. S414R is a major factor in amoxicillin resistance in vitro [29]. However, the latter substitution was detected here in two amoxicillin-sensitive strains with MICs of 0.06 and 0.12, which was higher than the average MIC of sensitive strains (0.03 μg/mL). Therefore, whether S414R alone contributes to the development of amoxicillin resistance requires further study.

Although previous studies prove that mutations in or adjacent to the PBP motifs confer amoxicillin resistance [24,31], we discovered the unique nonsense mutation at codon L610*, which is located far from the third PBP-motif KTG_555-7_ because of an upstream stop codon, which may confer amoxicillin resistance with an additional substitution such as S414R. We did not detect mutations in *pbp1A* in two amoxicillin-resistant strains, which may be explained by mutations in penicillin-binding proteins 2 and 3 [32]. Amoxicillin resistance was significantly higher in our present study and was associated with several previously unidentified mutations. We are therefore attempting to investigate the mechanism.

Clarithromycin is crucial for the eradication of *H. pylori* [2]. However, a recent meta-analysis found higher clarithromycin-resistance rates throughout the Asian-Pacific region (approximately 34% in Vietnam, 19% in Japan, and 26% in China, overall resistance rate increased from 7% to 21% during the last two decades [11]. In Mongolia, the higher clarithromycin-resistance rate (29.9%) compared with the overall resistance rate (>15%) in all study areas is an alarming sign (Table 2). The Maastricht V guidelines recommend avoid administering clarithromycin in regions where resistance exceeds 15% [2].

Our present results are consistent with those of a study of drug resistance in Ulaanbaatar [22,27]. In our recent clinical trial conducted in Ulaanbaatar, a susceptibility test-based clarithromycin-triple therapy regimen achieved a >90% cure rate, as determined using Per Protocol analysis [27]. These findings suggest the importance of determining resistance rates and identifying the underlying genetic mechanisms. Here we determined the full sequence of the 23S rRNA gene, including the domain V, which harbors mutations frequently associated with resistance [10]. For example, we found that the known mutation A2143G in domain V was significantly associated with clarithromycin resistance. Furthermore, the most frequent (53–95%) mutation was A2143G, followed by A2142G and A2142C [10]. However, the latter two mutations were not detected here, and five strains carried the A2143G mutation with unknown substitutions. Two or more mutations in the 23S rRNA may be the result of previous exposure to macrolides [33]. Moreover, the *H. pylori* strains analyzed here carried new nucleotide substitutions in or outside of domain V as follows: A1410G, C1707T, A2167G, C2248T+G2287A, and C2922T. The G2287A mutation was detected in Vietnam [34]. Further studies are therefore required to explain the variety of antibiotic-resistance determinants in the 23S rRNA gene.

The resistance rate to metronidazole varies according to geographical region and is mainly detected at a constant high level, particularly in economically disadvantaged countries [19]. Furthermore, in Asian-Pacific countries, metronidazole resistance increased from 36% to 45% during the last 20 years [11]. Our present study reveals an extremely high rate of resistance to metronidazole (overall 78.7%) and that woman have a higher resistance rate (≤82.6%) than men (Table 1). Higher resistance is caused by the widespread use of the drug to treat non-*H. pylori* diseases such as oral and gynecological disorders [10]. Metronidazole resistance is conferred by alterations of *rdxA* and *frxA* [17], and the 96.6% (56/58) rate of resistance to metronidazole can be explained by mutations in these genes. In the present study, we show that both genes harbor a high frequency of missense mutations (46.8% and 39%, respectively).

Most *frxA* genes sequenced here possessed frameshift that generated an early stop codon. In contrast, 20.6% (12/58) of metronidazole-resistant strains did not carry a mutation in *rdxA*. These findings suggest that *frxA* is required for the development of metronidazole resistance, at least in Mongolia. Independent of *rdxA*, truncation of *frxA* contributes to metronidazole resistance [35]. A recent meta-analysis found that *H. pylori* is resistant to metronidazole and clarithromycin, which is associated with a <80% decrease in the efficacy of triple therapy using either antibiotic [36].

A fluoroquinolone (levofloxacin)-based regimen is recommended for salvage therapy for *H. pylori* eradication, after failure of first-line therapy [2,37]. Fluoroquinolone resistance differs according to its use worldwide, and high resistance rates to levofloxacin are present in China (28%), Turkey (28%), and Cambodia (67.3%) [11,38,39]. We found here that the overall levofloxacin resistance rate was 41.3% in Mongolia, which is consistent with studies of these countries. The mechanism of resistance to levofloxacin involves mutations in the QRDR of GyrA [26]. In the present study, 60% (15/25) of the levofloxacin-resistant strains carried mutations at D91G/N/Y. Interestingly, most strains with a codon change at Asn 87 Lys had higher MICs (8–16 μg/mL). Further, we detected the A88P substitution in GyrA in one resistant strain. Two previous studies found this rare mutation in a strain with MIC = 2 mg/L, among 97 fluoroquinolone-resistant isolates [40], and the other reported the novel double mutations N87 and A88 in *H. pylori* resistant to high concentrations of sitafloxacin [41]. Here, despite the known mutations of codons N87 and D91, the single mutation A88P conferred a high level of levofloxacin resistance upon one of 74 Mongolian strains.

Generally, the rates of resistance to tetracycline and its derivatives resistance are <10%, except for the Eastern Mediterranean and African countries [19]. Here, we found that one strain among 361 (0.28%) was resistant to minocycline (MIC = 2 μg/mL) and carried a mutation of codon A928C. Minocycline binds to the 30S subunit of the bacterial 70S ribosome, which inhibits protein synthesis. The half-life of minocycline is longer than that of tetracycline, and the drug is more lipid-soluble, which enhances its infiltration into tissue [42]. For example, minocycline-based triple therapy is effective for second-line therapy of *H. pylori* infection [42,43], indicating that a regimen including tetracycline or minocycline could serve as second- or third-line therapy in Mongolia. Our present study found a high incidence (51.3%) of MDR among *H. pylori* isolated in Mongolia, suggesting that minocycline may represent an important candidate as a treatment option. MDR is the main obstacle that prevents achieving an acceptable eradication rate because MDR profiles comprise first-line therapy components.

## 5. Conclusions

This is the first study to investigate the antibiotic susceptibility rate of clinical isolates of *H. pylori* and the associated bacterial genetic determinants across Mongolia. The rate of antibiotic resistance of *H. pylori* infections is high, particularly to amoxicillin, clarithromycin, metronidazole, and levofloxacin, which indicates the insufficient efficiency of standard triple therapy. Therefore, the present study serves as a foundation for developing national guidelines for more effective therapeutic strategies. Moreover, it is essential to conduct appropriate analyses using NGS to identify the genetic determinants of antibiotic resistance before treatment. Thus, we show here that the causes of antibiotic resistance were associated with known mutations. Moreover, we discovered novel mutations in the *pbp1A* that encodes amoxicillin resistance as well as in the 23S rRNA gene of clarithromycin-resistant isolates. NGS provides a comprehensive and powerful tool for determining antibiotic-resistance associated with genetic determinants.

## Figures and Tables

**Figure 1 microorganisms-08-01062-f001:**
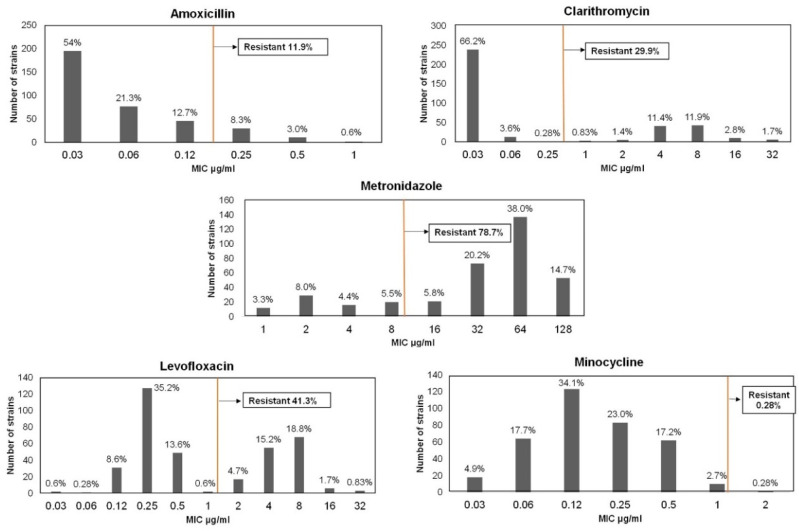
Antibiotic Resistance Rates. Outcomes according to agar dilution assays of 361 *H. pylori* isolates collected from five different locations in Mongolia. Clinical breakpoints were defined according to the guidelines of the European Committee on Antimicrobial Susceptibility Testing (EUCAST) (orange vertical line).

**Figure 2 microorganisms-08-01062-f002:**
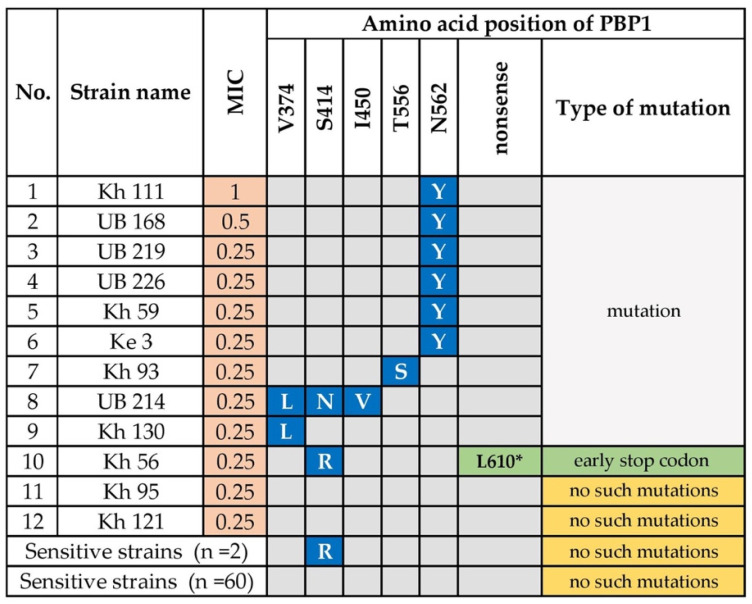
Amino Acid Substitutions in the Penicillin-binding Protein 1 of Amoxicillin-Resistant *H. pylori*. N; asparagine; T; threonine; V; valine; S; serine; I; isoleucine; Y; tyrosine; L; leucine; R; arginine. Amino acid substitution highlighted in blue (Y; S; L; N; V) were only detected in the amoxicillin-resistant strains. The amino acid substitution Arg (R) was detected in two amoxicillin-sensitive strains. The L610* mutation highlighted in green indicates a possible novel mutation. Kh, Khuvsgul Province; UB, Ulaanbaatar city; Ke, Khentii Province.

**Figure 3 microorganisms-08-01062-f003:**
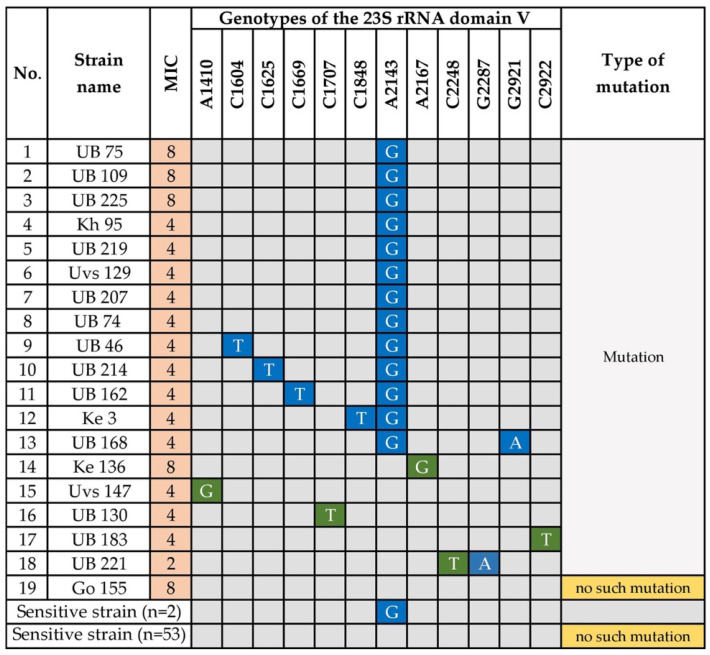
Point Mutations in 23s rRNA Genes of Clarithromycin-Resistant H. pylori. A; Adenine, G; Guanine, C; Cytosine, T; Thymine. Blue indicates the presence of the corresponding mutation. Nucleotides highlighted in green are novel mutations. UB, Ulaanbaatar city. Provinces: Kh, Khuvsgul; Uvs, Uvs; Ke, Khentii; Go, Umnugovi.

**Figure 4 microorganisms-08-01062-f004:**
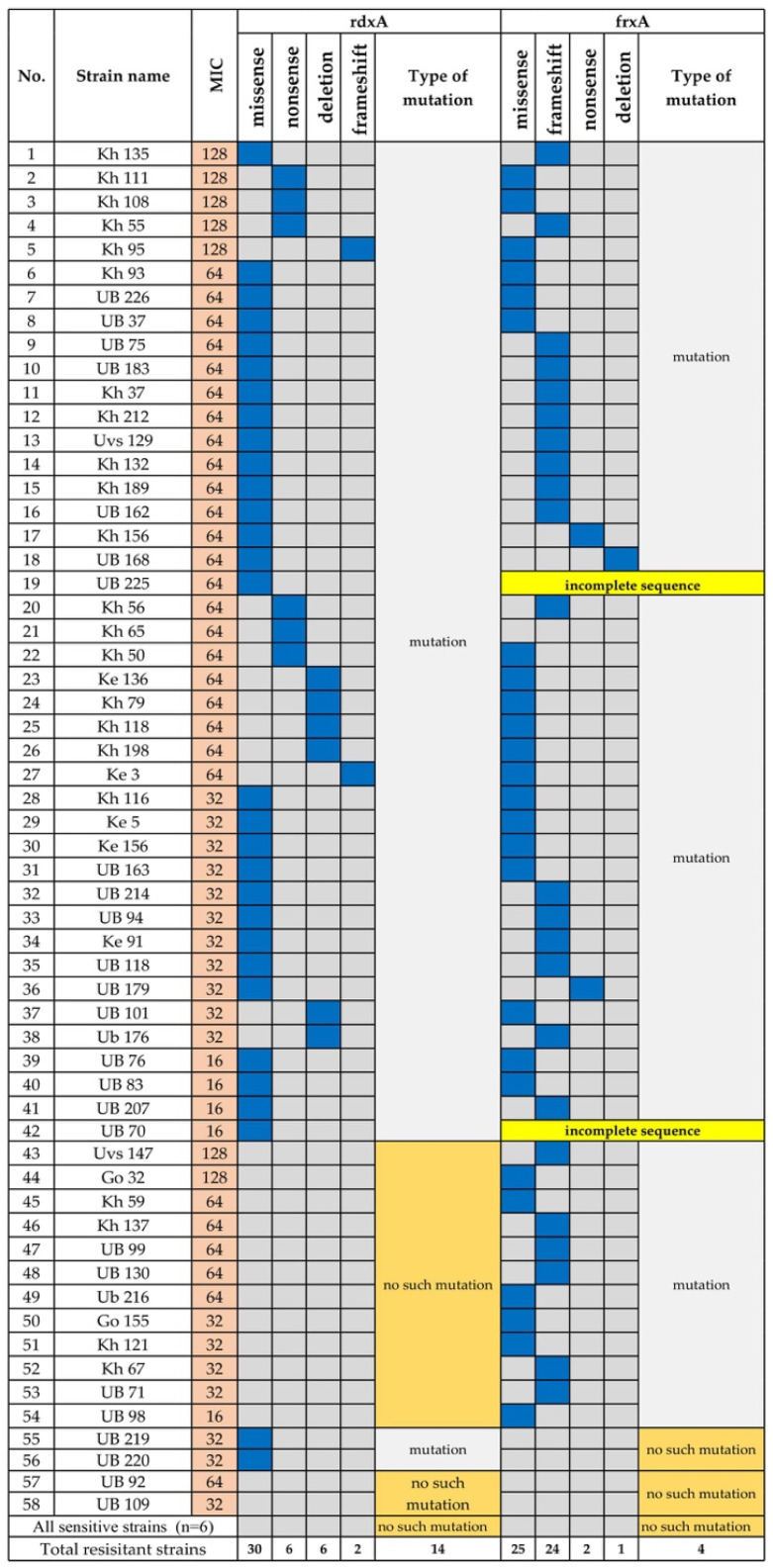
Mutations in *rdxA* and *frxA* of Metronidazole-Resistant *H. pylori*. Blue indicates the corresponding mutation. UB, Ulaanbaatar city. Provinces: Kh, Khuvsgul; Uvs, Uvs; Ke, Khentii; Go, Umnugovi.

**Figure 5 microorganisms-08-01062-f005:**
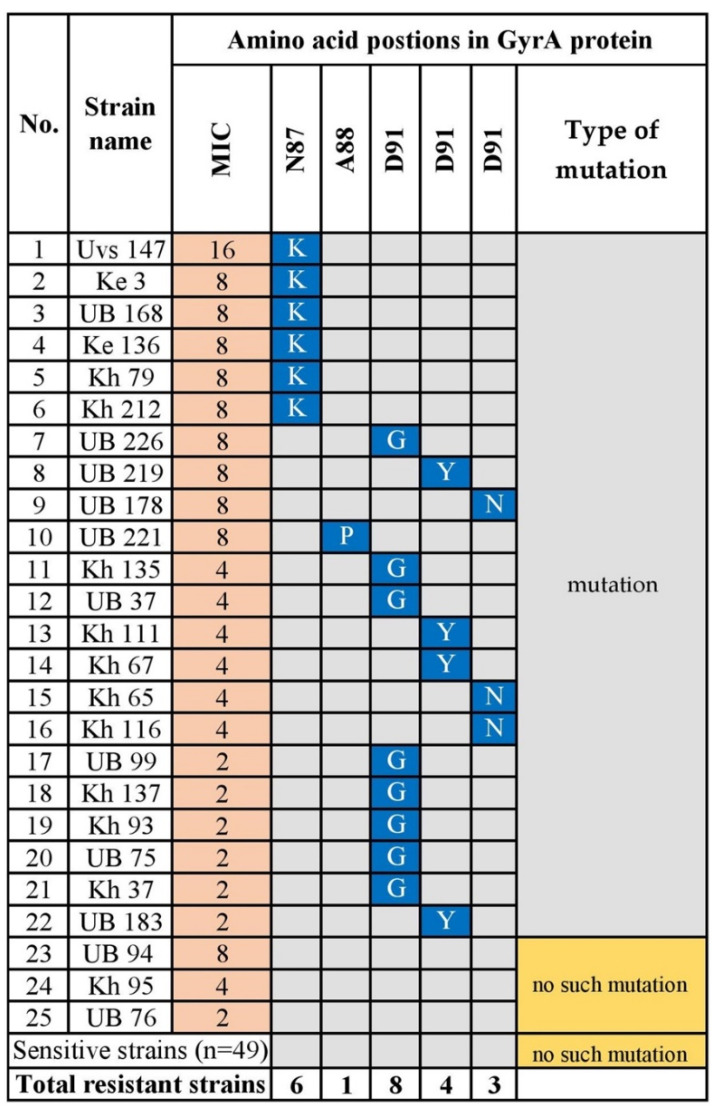
GyrA Amino Acid Substitutions Associated with Levofloxacin-Resistant *H. pylori*. N; asparagine; A; alanine; Y; tyrosine; D; Aspartic acid; K; lysine; P; proline; G; glycine. Blue indicates the corresponding mutation. UB, Ulaanbaatar city. Provinces: Kh, Khuvsgul; Uvs, Uvs; Ke, Khentii.

**Table 1 microorganisms-08-01062-t001:** Demographics and Antibiotic Resistance Rates.

Group	Total, *n* (%)	Antibiotic-Resistant Strains, *n* (%)				
		Amoxicillin	Clarithromycin	Metronidazole	Levofloxacin	Minocycline
Total, *n*	361	43 (11.9)	108 (29.9)	284 (78.7)	149 (41.3)	1 (0.28)
**Sex**						
Female	264 (73.1)	31 (11.7)	85 (32.2)	218 (82.6) **	113 (42.8)	0 (0)
Male	97 (26.9)	12 (12.4)	23 (23.7)	66 (68.0)	36 (37.1)	1 (1.0)
Age ^#^						
19–29	65	9 (13.8)	20 (30.8)	44 (67.7)	21 (32.3)	0 (0)
30–39	77	7 (9.1)	27 (35.1)	66 (85.7)	34 (44.2)	1 (1.3)
40–49	78	6 (7.7)	23 (29.5)	63 (80.8)	34 (43.6)	0 (0)
50–59	91	13 (14.3)	25 (27.5)	75 (82.4)	40 (44.0)	0 (0)
60<	50	8 (16.0)	13 (26.0)	36 (72.0)	20 (40.0)	0 (0)

** *p = 0.003*, ^#^ 44.3 ± 13.4 years of age (mean ± SD).

**Table 2 microorganisms-08-01062-t002:** Geographical Distribution of Antibiotic Resistance Rates.

Areas	Total, *n*	Amoxicillin		Clarithromycin		Metronidazole		Levofloxacin		Minocycline	
		*n*	% (95% CI)	*n*	% (95% CI)	*n*	% (95% CI)	*n*	% (95% CI)	*n*	% (95% CI)
**Khentii**	90	12	13.3 (7.5–21.5)	22	24.4 (16.5–34)	75	83.3 (74.6–89.9)	42	46.7 (36.6–56.9)	0	0.0
**Khuvsgul**	35	7	20.0 (9.4–35.3)	6	17.1 (7.5–32)	29	82.9 (68–92.5)	14	40.0 (25.1–56.5)	0	0.0
**Ulaanbaatar**	124	11	8.9 (4.8–14.8)	42	33.9 (26–42.5)	91	73.4 (65.1–80.6)	47	37.9 (29.7–46.6)	1	0.8 (0.1–3.7)
**Umnugovi**	84	6	7.1 (3.0–14.1)	24	28.6 (19.8–38.8)	68	81.0 (71.6–88.2)	35	41.7 (31.6–52.3)	0	0.0
**Uvs**	28	7	**25.0** * (11.9–42.9)	14	**50.0** * (32.2–67.8)	21	75.0 (57.1–88.1)	11	39.3 (23–57.7)	0	0.0
		* *p* = 0.043		* *p* = 0.034		NS		NS		NS	

NS; not significant.

**Table 3 microorganisms-08-01062-t003:** Multidrug Resistance Patterns.

Resistance Pattern	Number of Strains	Percentage (95% CI)
Sensitive to All Antibiotics	43	11.9% (8.9–15.6)
**Single drug**	133	
MNZ	106	29.4% (24.8–34.2)
LEV	11	3.0% (1.6–5.2)
CLR	10	2.8% (1.4–4.9)
AMX	5	1.4% (0.5–3.0)
MNO	1	0.28% (0.0–1.3)
**Two drugs**	113	
MNZ + LEV	66	18.3% (14.6–22.5)
CLR + MNZ	33	9.1% (6.5–12.4)
AMX + MNZ	9	2.5% (1.2–4.5)
CLR + LEV	5	1.4% (0.5–3.0)
**Three drugs**	62	
CLR + MNZ + LEV	43	11.9% (8.9–15.6)
AMX + MNZ + LEV	12	3.3% (1.8–5.6)
AMX + CLR + MNZ	5	1.4% (0.5–3.0)
AMX + CLR + LEV	2	0.56% (0.1–1.8)
**Four drugs**	10	
AMX+CLR+MNZ+LEV	10	2.8% (1.4–4.9)

Abbreviations: AMX, amoxicillin; CLR, clarithromycin; MNZ, metronidazole; LEV, levofloxacin; MNO, minocycline.

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
