# Peer review of "High Antibiotic Resistance of Helicobacter pylori and Its Associated Novel Gene Mutations among the Mongolian Population"

_microorganisms, 2020, doi:10.3390/microorganisms8071062_

Round 1
Reviewer 1 Report
The purpose of the “High Antibiotic Resistance for Helicobacter pylori and Its Associated Novel Gene Mutations among Mongolian Population“ article was to show the prevalence of antibiotic resistant H. pylori strains and the presence of genetic mutations responsible for their development (including novel ones) among Mongolian population.
While studies dealing with the topic of resistance in H. pylori are relatively often undertaken, the results presented in this article may arouse a scientific and clinical interest.
The article is written in a very good way, with the correct language. The number of mistakes in the text is small. Suggested corrections:
- Mongolia has the high prevalence of Helicobacter pylori infection and the second highest incidence of gastric cancer in the world. -> Mongolia has the high prevalence of Helicobacter pylori infection and the second highest incidence of gastric cancers in the world. [Abstract, line 19-20]
- Thus, investigating antibiotic resistance prevalence and its underlying genetic mechanism is necessary. [Abstract, line 20]
- Agar dilution assay was used to determine the minimum inhibition concentration of five antibiotics. -> please name them [Abstract, line 23]
- Antibiotic resistance rate to H. pylori infection was high, indicates that inefficient result of standard triple therapy in Mongolia. -> Antibiotic resistance rate of pylori was high, indicating that inefficient result of standard triple therapy in Mongolia. [Abstract, line 30-31]
- … levofloxacin and did not take into the genetic mechanisms of antibiotic resistance deeply. -> … levofloxacin and did not take into consideration the genetic mechanisms of antibiotic resistance deeply [Introduction, line 74]
- … minocycline, which is the second generation of tetracycline derivatives and their related gene mutations from different areas in Mongolia. -> … minocycline, which is the second generation of tetracycline derivatives, and their related gene mutations from different areas in Mongolia. [Introduction, line 79]
- Detection of pylori antibiotic resistance mutation based on the molecular approach can provide -> Detection of H. pylori antibiotic resistance mutations based on the molecular approach can provide [Introduction, line 80]
- Patients aged more than 16years old who were willing to perform upper -> Patients aged more than 16 years old who were willing to perform upper [Materials and Methods, line 86]
- Intriguingly we found newly the strain (Kh 130) possessed S414R with a unique early stop codon at 610. -> Intriguingly, we found newly the strain (Kh 130) possessed S414R with a unique early stop codon at 610. [Discussion, line 337]
- Another antibiotic used for the standard triple therapy, clarithromycin is crucial for the successful eradication of pylori [2]. -> Another antibiotic used for the standard triple therapy, clarithromycin, is crucial for the successful eradication of H. pylori [2]. [Discussion, line 352]
- Kim JM et al. reported that dual or more mutations in 23S rRNA might be result of previous exposure to macrolide [33]. -> Kim JM et al. reported that dual or more mutations in 23S rRNA might be result of previous exposure to macrolides [33]. [Discussion, line 370]
- Fluoroquinolone, levofloxacin-based regimen is recommended for salvage therapy for pylori eradication after first-line therapy failure [2, 37]. -> Fluoroquinolone (levofloxacin)-based regimen is recommended for salvage therapy for H. pylori eradication after first-line therapy failure [2, 37]. [Discussion, linen 391]
Author Response
Re: [Microorganisms] Manuscript ID: microorganisms-843108 – Minor Revisions
Dear Reviewer 1,
We thank you for the comments made on the Manuscript ID: microorganisms-843108. We are submitting herewith a revised version of the manuscript and a point-by-point response to the issues kindly raised by your comments.
We hope that the above revisions have fulfilled your expectations and thank you for your revision work.
Yoshio Yamaoka on behalf of the authors
Responses to comments
General comments:
-
The purpose of the “High Antibiotic Resistance for Helicobacter pylori and Its Associated Novel Gene Mutations among Mongolian Population“ article was to show the prevalence of antibiotic resistant H. pylori strains and the presence of genetic mutations responsible for their development (including novel ones) among Mongolian population. While studies dealing with the topic of resistance in H. pylori are relatively often undertaken, the results presented in this article may arouse a scientific and clinical interest. The article is written in a very good way, with the correct language. The number of mistakes in the text is small. Answer: We thank the reviewer for the interest toward our study and proper observations.
Specific comments:
-
Abstract: Mongolia has the high prevalence of Helicobacter pylori infection and the second highest incidence of gastric cancer in the world. -> Mongolia has the high prevalence of Helicobacter pylori infection and the second highest incidence of gastric cancers in the world. [Abstract, line 19-20]. Answer: We corrected as pointed out (page 1, line 20).
-
Thus, investigating antibiotic resistance prevalence and its underlying genetic mechanism is necessary. [Abstract, line 20] . Answer: We corrected as pointed out, and also revised as “the prevalence of antibiotic” (page 1, line 21).
-
Agar dilution assay was used to determine the minimum inhibition concentration of five antibiotics. -> please name them [Abstract, line 23]. Answer: We added the name of the antibiotics as suggested (page 1, lines 24-25).
-
Antibiotic resistance rate to H. pylori infection was high, indicates that inefficient result of standard triple therapy in Mongolia. -> Antibiotic resistance rate to H. pylori was high, indicating that inefficient result of standard triple therapy in Mongolia. [Abstract, line 30-31]. Answer: We have changed the sentence as suggested (page 1, lines 32-33).
-
Introduction: … levofloxacin and did not take into the genetic mechanisms of antibiotic resistance deeply. -> … levofloxacin and did not take into consideration the genetic mechanisms of antibiotic resistance deeply [Introduction, line 74]. Answer: We have changed the sentence as suggested (page 2, line 75).
-
… minocycline, which is the second generation of tetracycline derivatives and their related gene mutations from different areas in Mongolia. -> … minocycline, which is the second generation of tetracycline derivatives, and their related gene mutations from different areas in Mongolia. [Introduction, line 79]. Answer: As advised, comma has been added after tetracycline derivatives (page 2, line 80-82).
-
Detection of H. pylori antibiotic resistance mutation based on the molecular approach can provide -> Detection of H. pylori antibiotic resistance mutations based on the molecular approach can provide [Introduction, line 80]. Answer: We corrected as pointed out (page 2, line 81-82).
-
Materials and Methods Patients aged more than 16years old who were willing to perform upper -> Patients aged more than 16 years old who were willing to perform upper [Materials and Methods, line 86]. Answer: We corrected as pointed out (page 2, lines 90).
-
Discussion Intriguingly we found newly the strain (Kh 130) possessed S414R with a unique early stop codon at 610. -> Intriguingly, we found newly the strain (Kh 130) possessed S414R with a unique early stop codon at 610. [Discussion, line 337]. Answer: We corrected as pointed out (page 12, lines 359-360).
-
Another antibiotic used for the standard triple therapy, clarithromycin is crucial for the successful eradication of pylori [2]. -> Another antibiotic used for the standard triple therapy, clarithromycin, is crucial for the successful eradication of H. pylori [2]. [Discussion, line 352]. Answer: We corrected as pointed out (page 12, lines 374-375).
-
Kim JM et al. reported that dual or more mutations in 23S rRNA might be result of previous exposure to macrolide [33]. -> Kim JM et al. reported that dual or more mutations in 23S rRNA might be result of previous exposure to macrolides [33]. [Discussion, line 370]. Answer: We corrected as pointed out (page 13, lines 391-393).
-
Fluoroquinolone, levofloxacin-based regimen is recommended for salvage therapy for pylori eradication after first-line therapy failure [2, 37]. -> Fluoroquinolone (levofloxacin)-based regimen is recommended for salvage therapy for H. pylori eradication after first-line therapy failure [2, 37]. [Discussion, linen 391]. Answer: We changed as pointed out (page 13, line 413-414).

Reviewer 2 Report
The paper investigates the prevalence of antibiotic resistance and its underlying genetic mechanism by isolating 361 H. pylori strains from different locations of Mongolia. Antibiotic resistance and the related genetic causes has been evaluated through next generation sequencing. Findings suggest that antibiotic resistance rate to H. pylori infection in Mongolia is high.
As the Ethical issue paragraph (2.4) is separated, I would delete the word ‘’Ethics’’ in 2.1. Study Population, Sampling and Ethics paragraph.
Statistics are properly conducted. The Materials and methods section is quite clear, as well as the results description. Figures 2,3,4 and 5 could be improved in their quality.
The final results support the conclusion of the paper.
Some typing errors must be reviewed throughout the manuscript.
Author Response
Re: [Microorganisms] Manuscript ID: microorganisms-843108 – Minor Revisions
Dear Reviewer 2,
We thank you for the comments made on the Manuscript ID: microorganisms-843108. We are submitting herewith a revised version of the manuscript and a point-by-point response to the issues kindly raised by your comments.
We hope that the above revisions have fulfilled your expectations and thank you for your revision work.
Yoshio Yamaoka on behalf of the authors
Responses to comments
General comments:
-
The paper investigates the prevalence of antibiotic resistance and its underlying genetic mechanism by isolating 361 H. pylori strains from different locations of Mongolia. Antibiotic resistance and the related genetic causes has been evaluated through next generation sequencing. Findings suggest that antibiotic resistance rate to H. pylori infection in Mongolia is high. Answer: We are grateful for the reviewer’s encouragement and positive comments on the manuscript.
Specific comments:
-
As the Ethical issue paragraph (2.4) is separated, I would delete the word ‘’Ethics’’ in 2.1. Study Population, Sampling and Ethics paragraph. Answer: Thank you very much for your comment. We deleted “Ethic” in 2.1 as suggested (page 2, line 87).
-
Statistics are properly conducted. The Materials and methods section is quite clear, as well as the results description. Figures 2, 3, 4 and 5 could be improved in their quality. Answer: The Figures have been improved as stated in the submission guideline, minimum 1000 pixels width/height, or a resolution of 300 dpi or higher.
-
The final results support the conclusion of the paper. Answer: We thank for positive and impressive comments on the manuscript.
-
Some typing errors must be reviewed throughout the manuscript. Answer: We have checked the manuscript carefully, and the typing errors and English grammar errors had been corrected (please see blue color letter in the text).
